# Diversity-sensitive Conditional Generative Adversarial Networks

**Dingdong Yang**[*][†]**, Seunghoon Hong**[*][†]**, Yunseok Jang**[†]**, Tianchen Zhao**[†]**, Honglak Lee**[†][‡]
[†] University of Michigan, Ann Arbor, MI, USA
[‡] Google Brain, Mountain View, CA, USA

## Abstract

We propose a simple yet highly effective method that addresses the mode-collapse problem in the Conditional Generative Adversarial Network (cGAN). Although conditional distributions are multi-modal (i.e., having many modes) in practice, most cGAN approaches tend to learn an overly simplified distribution where an input is always mapped to a single output regardless of variations in latent code. To address such issue, we propose to explicitly regularize the generator to produce diverse outputs depending on latent codes. The proposed regularization is simple, general, and can be easily integrated into most conditional GAN objectives. Additionally, explicit regularization on generator allows our method to control a balance between visual quality and diversity. We demonstrate the effectiveness of our method on three conditional generation tasks: image-to-image translation, image inpainting, and future video prediction. We show that simple addition of our regularization to existing models leads to surprisingly diverse generations, substantially outperforming the previous approaches for multi-modal conditional generation specifically designed in each individual task.

## 1 Introduction

The objective of conditional generative models is learning a mapping function from input to output distributions. Since many conditional distributions are inherently ambiguous (*e.g.* predicting the future of a video from past observations), the ideal generative model should be able to learn a multi-modal mapping from inputs to outputs. Recently, Conditional Generative Adversarial Networks (cGAN) have been successfully applied to a wide-range of conditional generation tasks, such as image-to-image translation (Isola et al., 2017; Wang et al., 2018; Zhu et al., 2017a), image inpainting (Pathak et al., 2016; Iizuka et al., 2017), text-to-image synthesis (Huang et al., 2017; Hong et al., 2018), video generation (Villegas et al., 2017), *etc.*. In conditional GAN, the generator learns a deterministic mapping from input to output distributions, where the multi-modal nature of the mapping is handled by sampling random latent codes from a prior distribution.

However, it has been widely observed that conditional GANs are often suffered from the mode collapse problem (Salimans et al., 2016; Arjovsky & Bottou, 2017), where only small subsets of output distribution are represented by the generator. The problem is especially prevalent for high-dimensional input and output, such as images and videos, since the model is likely to observe only one example of input and output pair during training. To resolve such issue, there has been recent attempts to learn multi-modal mapping in conditional generative models (Zhu et al., 2017b; Huang et al., 2018). However, they are focused on specific conditional generation tasks (*e.g.* image-to-image translation) and require specific network architectures and objective functions that sometimes are not easy to incorporate into the existing conditional GANs.

In this work, we introduce a simple method to regularize the generator in conditional GAN to resolve the mode-collapse problem. Our method is motivated from an observation that the mode-collapse happens when the generator maps a large portion of latent codes to similar outputs. To avoid this, we propose to encourage the generator to produce different outputs depending on the latent code, so as to learn a one-to-one mapping from the latent codes to outputs instead of many-to-one. Despite the simplicity, we show that the proposed method is widely applicable to various cGAN architectures and tasks, and outperforms more complicated methods proposed to achieve multi-modal conditional generation for specific tasks. Additionally, we show that we can control a balance between visual

---

[*]Equal contribution

quality and diversity of generator outputs with the proposed formulation. We demonstrate the effectiveness of the proposed method in three representative conditional generation tasks, where most existing cGAN approaches produces deterministic outputs: *Image-to-image translation, image inpainting* and *video prediction*. We show that simple addition of the proposed regularization to the existing cGAN models effectively induces stochasticity from the generator outputs.

## 2 RELATED WORK

Resolving the mode-collapse problem in GAN is an important research problem, and has been extensively studied in the standard GAN settings (Metz et al., 2017; Arjovsky et al., 2017; Gulrajani et al., 2017; Salimans et al., 2016; Miyato et al., 2018). These approaches include unrolling the generator gradient update steps (Metz et al., 2017), incorporating the minibatch statistics into the discriminator (Salimans et al., 2016), employing the improved divergence measure to smooth the loss landscape of the discriminator (Gulrajani et al., 2017; Arjovsky et al., 2017; Miyato et al., 2018), *etc.*. Although these approaches have been successful in modeling unconditional data distribution to some extent, recent studies have reported that it is still not sufficient to resolve a mode-collapse problem in many conditional generative tasks, especially for high-dimensional input and output.

Recently, some approaches have been proposed to address the mode-collapse issue in conditional GAN. Zhu et al. (2017b) proposed a hybrid model of conditional GAN and Variational Autoencoder (VAE) for multi-modal image-to-image translation task. The main idea is designing the generator to be invertible by employing an additional encoder network that predicts the latent code from the generated image. The similar idea has been applied to unsupervised image-to-image translation (Huang et al., 2018) and stochastic video generation (Lee et al., 2018) but with non-trivial task-specific modifications. However, these approaches are designed to achieve multi-modal generation in each specific task, and there has been no unified solution that addresses the mode-collapse problem for general conditional GANs. Recently, (Odena et al., 2018) proposed a method that regularizes the generator by clamping the generator Jacobian within a certain range. Our method shares the similar motivation with (Odena et al., 2018) but employs a different objective function that simply maximizes the norm of the generator gradient with an optional upper-bound, which we found that works much more stable over a wide range of tasks with less number of hyper-parameters.

## 3 METHOD

Consider a problem of learning a conditional mapping function $G : \mathcal{X} \rightarrow \mathcal{Y}$, which generates an output $\mathbf{y} \in \mathcal{Y}$ conditioned on the input $\mathbf{x} \in \mathcal{X}$. Our goal is to learn a multi-modal mapping $G : \mathcal{X} \times \mathcal{Z} \rightarrow \mathcal{Y}$, such that an input $\mathbf{x}$ can be mapped to multiple and diverse outputs in $\mathcal{Y}$ depending on the latent factors encoded in $\mathbf{z} \in \mathcal{Z}$. To learn such multi-modal mapping $G$, we consider a conditional Generative Adversarial Network (cGAN), which learns both conditional generator $G$ and discriminator $D$ by optimizing the following adversarial objective:

$$\min_G \max_D \mathcal{L}_{cGAN}(G, D) = \mathbb{E}_{\mathbf{x},\mathbf{y}}[\log D(\mathbf{x}, \mathbf{y})] + \mathbb{E}_{\mathbf{x},\mathbf{z}}[\log(1 - D(\mathbf{x}, G(\mathbf{x}, \mathbf{z})))]. \quad (1)$$

Although conditional GAN has been proved to work well for many conditional generation tasks, it has been also reported that optimization of Eq. (1) often suffers from the mode-collapse problem, which in extreme cases leads the generator to learn a deterministic mapping from $\mathbf{x}$ to $\mathbf{y}$ and ignore any stochasticity induced by $\mathbf{z}$. To address such issue, previous approaches encouraged the generator to learn an invertible mapping from latent code to output by $E(G(\mathbf{x}, \mathbf{z})) = \mathbf{z}$ (Zhu et al., 2017b; Huang et al., 2018). However, incorporating an extra encoding network $E$ into the existing conditional GANs requires non-trivial modification of network architecture and introduce the new training challenges, which limits its applicability to various models and tasks.

We introduce a simple yet effective regularization on the generator that directly penalizes its mode-collapsing behavior. Specifically, we add the following *maximization* objective to the generator:

$$\max_G \mathcal{L}_{\mathbf{z}}(G) = \mathbb{E}_{\mathbf{z}_1,\mathbf{z}_2}\left[\min\left(\frac{\|G(\mathbf{x}, \mathbf{z}_1) - G(\mathbf{x}, \mathbf{z}_2)\|}{\|\mathbf{z}_1 - \mathbf{z}_2\|}, \tau\right)\right], \quad (2)$$

where $\|\cdot\|$ indicates a norm and $\tau$ is a bound for ensuring numerical stability. The intuition behind the proposed regularization is very simple: when the generator collapses into a single mode and produces deterministic outputs based only on the conditioning variable $\mathbf{x}$, Eq. (2) approaches its minimum since $G(\mathbf{x}, \mathbf{z}_1) \approx G(\mathbf{x}, \mathbf{z}_2)$ for all $\mathbf{z}_1, \mathbf{z}_2 \sim N(\mathbf{0}, \mathbf{1})$. By regularizing generator to maximize Eq. (2), we force the generator to produce diverse outputs depending on latent code $\mathbf{z}$.

Our full objective function can be written as:

$$\min_G \max_D \ \mathcal{L}_{cGAN}(G, D) - \lambda \mathcal{L}_{\boldsymbol{z}}(G), \tag{3}$$

where $\lambda$ controls an importance of the regularization, thus, the degree of stochasticity in $G$. If $G$ has bounded outputs through a non-linear output function (*e.g.* sigmoid), we remove the margin from Eq. (2) in practice and control its importance only with $\lambda$. In this case, adding our regularization introduces only one additional hyper-parameter.

The proposed regularization is simple, general, and can be easily integrated into most existing conditional GAN objectives. In the experiment, we show that our method can be applied to various models under different objective functions, network architectures, and tasks. In addition, our regularization allows an explicit control over a degree of diversity via hyper-parameter $\lambda$. We show that different types of diversity emerge with different $\lambda$. Finally, the proposed regularization can be extended to incorporate different distance metrics to measure the diversity of samples. We show this extension using distance in feature space and for sequence data.

## 4 ANALYSIS OF THE PROPOSED REGULARIZATION

**Connection to Generator Gradient.** We show in Appendix A that the proposed regularization in Eq. (2) corresponds to a lower-bound of averaged gradient norm of $G$ over $[\boldsymbol{z}_1, \boldsymbol{z}_2]$ as:

$$\mathbb{E}_{\boldsymbol{z}_1, \boldsymbol{z}_2} \left[ \frac{\|G(\boldsymbol{x}, \boldsymbol{z}_2) - G(\boldsymbol{x}, \boldsymbol{z}_1)\|}{\|\boldsymbol{z}_2 - \boldsymbol{z}_1\|} \right] \leq \mathbb{E}_{\boldsymbol{z}_1, \boldsymbol{z}_2} \left[ \int_0^1 \|\nabla_{\boldsymbol{z}} G(\boldsymbol{x}, \boldsymbol{\gamma}(t))\| dt \right] \tag{4}$$

where $\boldsymbol{\gamma}(t) = t\boldsymbol{z}_2 + (1 - t)\boldsymbol{z}_1$ is a straight line connecting $\boldsymbol{z}_1$ and $\boldsymbol{z}_2$. It implies that optimizing our regularization (LHS of Eq. (4)) will increase the gradient norm of the generator $\|\nabla_{\boldsymbol{z}} G\|$.

It has been known that the GAN suffers from a gradient vanishing issue (Arjovsky & Bottou, 2017) since the gradient of optimal discriminator vanishes almost everywhere $\nabla D \approx 0$ except near the true data points. To avoid this issue, many previous works had been dedicated to smoothing out the loss landscape of $D$ so as to relax the vanishing gradient problem (Arjovsky et al., 2017; Miyato et al., 2018; Gulrajani et al., 2017; Kurach et al., 2018). Instead of smoothing $\nabla D$ by regularizing discriminator, we increase $\|\nabla_{\boldsymbol{z}} G\|$ to encourage $G(\boldsymbol{x}, \boldsymbol{z})$ to be more spread over the output space from the fixed $\boldsymbol{z}_j \sim p(\boldsymbol{z})$, so as to capture more meaningful gradient from $D$.

**Optimization Perspective.** We provide another perspective to understand how the proposed method addresses the mode-collapse problem. For notational simplicity, here we omit the conditioning variable from the generator and focus on a mapping of latent code to output $G_\theta : \mathcal{Z} \to \mathcal{Y}$.

Let a mode $\mathcal{M}$ denotes a set of data points in an output space $\mathcal{Y}$, where all elements of the mode have very small differences that are perceptually indistinguishable. We consider that the mode-collapse happens if the generator maps a large portion of latent codes to the mode $\mathcal{M}$.

Under this definition, we are interested in a situation where the generator output $G_\theta(\boldsymbol{z}_1)$ for a certain latent code $\boldsymbol{z}_1$ moves closer to a mode $\mathcal{M}$ by a distance of $\epsilon$ via a single gradient update. Then we show in Appendix B that such gradient update at $\boldsymbol{z}_1$ will *also move* the generator outputs of neighbors in a neighborhood $\mathcal{N}_r(\boldsymbol{z}_1)$ to the same mode $\mathcal{M}$. In addition, the size of neighborhood $\mathcal{N}_r(\boldsymbol{z}_1)$ can be arbitrarily large but is bounded by an open ball of a radius $r = \epsilon \cdot \left( 4 \inf_{\boldsymbol{z}} \left\{ \max \left( \frac{\|G_{\theta_t}(\boldsymbol{z}_1) - G_{\theta_t}(\boldsymbol{z})\|}{\|\boldsymbol{z}_1 - \boldsymbol{z}\|}, \frac{\|G_{\theta_{t+1}}(\boldsymbol{z}_1) - G_{\theta_{t+1}}(\boldsymbol{z})\|}{\|\boldsymbol{z}_1 - \boldsymbol{z}\|} \right) \right\} \right)^{-1}$, where $\theta_t$ and $\theta_{t+1}$ denote the generator parameters before and after the gradient update, respectively.

Without any constraints on $\frac{\|G(\boldsymbol{z}_1) - G(\boldsymbol{z}_2)\|}{\|\boldsymbol{z}_1 - \boldsymbol{z}_2\|}$, a single gradient update can cause the generator outputs for a large amount of latent codes to be collapsed into a mode $\mathcal{M}$. We propose to shrink the size of such neighborhood by constraining $\frac{\|G(\boldsymbol{z}_1) - G(\boldsymbol{z}_2)\|}{\|\boldsymbol{z}_1 - \boldsymbol{z}_2\|}$ above some threshold $\tau > 0$, therefore prevent the generator placing a large probability mass around a mode $\mathcal{M}$.

**Connection with BicycleGAN (Zhu et al., 2017b).** We establish an interesting connection of our regularization with Zhu et al. (2017b). Recall that the objective of BicycleGAN is encouraging an invertibility of a generator by minimizing $\|\boldsymbol{z} - E(G(\boldsymbol{z}))\|_1$. By taking derivative with respect to $\boldsymbol{z}$, it implies that optimal $E$ will satisfy $I = \nabla_G E(G(\boldsymbol{z})) \nabla_{\boldsymbol{z}} G(\boldsymbol{z})$. Because an ideal encoder $E$ should be robust against spurious perturbations from inputs, we can naturally assume that the gradient norm of

$E$ should not be very large. Therefore, to maintain invertibility, we expect the gradient of $G$ should not be zero, i.e. $\|\nabla_{\boldsymbol{z}} G(\boldsymbol{z})\| > \tau$ for some $\tau > 0$, which prevents a gradient of the generator being vanishing. It is related to our idea that penalizes a vanishing gradient of the generator. Contrary to BicycleGAN, however, our method explicitly optimizes a generator gradient to have a reasonably high norm. It also allows us to *control* a degree of diversity with a hyper-parameter $\lambda$.

## 5 EXPERIMENTS

In this section, we demonstrate the effectiveness of the proposed regularization in three representative conditional generation tasks that most existing methods suffer from mode-collapse: *image-to-image translation, image inpainting* and *future frame prediction*. In each task, we choose an appropriate cGAN baseline from the previous literature, which produces realistic but deterministic outputs, and apply our method by simply adding our regularization to their objective function. We denote our method as DSGAN (**D**iversity-**S**ensitive **GAN**). Note that both cGAN and DSGAN use the exactly the same networks. Throughout the experiments, we use the following objective:

$$\min_{G} \max_{D} \; \mathcal{L}_{cGAN}(G, D) + \beta \mathcal{L}_{rec}(G) - \lambda \mathcal{L}_{\boldsymbol{z}}(G), \tag{5}$$

where $\mathcal{L}_{rec}(G)$ is a regression (or reconstruction) loss to ensure similarity between a prediction $\hat{\boldsymbol{y}}$ and ground-truth $\boldsymbol{y}$, which is chosen differently by each baseline method. Unless otherwise stated, we use $l_1$ distance for $\mathcal{L}_{rec}(G) = \|G(\boldsymbol{x}, \boldsymbol{z}) - \boldsymbol{y}\|$ and $l_1$ norm for $\mathcal{L}_{\boldsymbol{z}}(G)$[1]. We provide additional video results at anonymous website: https://sites.google.com/view/iclr19-dsgan/.

### 5.1 IMAGE-TO-IMAGE TRANSLATION

In this section, we consider a task of image-to-image translation. Given a set of training data $(\boldsymbol{x}, \boldsymbol{y}) \in (\mathcal{X}, \mathcal{Y})$, the objective of the task is learning a mapping $G$ that transforms an image in domain $\mathcal{X}$ to another image in domain $\mathcal{Y}$ (*e.g.* sketch to photo image).

As a baseline cGAN model, we employ the generator and discriminator architectures from BicycleGAN (Zhu et al., 2017b) for a fair comparison. We evaluate the results on three datasets: *label→image* (Radim Tyleček, 2013), *edge→photo* (Zhu et al., 2016; Yu & Grauman, 2014), *map→image* (Isola et al., 2017). For evaluation, we measure both the quality and the diversity of generation using two metrics from the previous literature. We employed Learned Perceptual Image Path Similarity (LPIPS) (Zhang et al., 2018) to measure the diversity of samples, which computes the distance between generated samples using features extracted from the pretrained CNN. Higher LPIPS score indicates more perceptual differences in generated images. In addition, we use Fréchet Inception Distance (FID) (Heusel et al., 2017) to measure the distance between training and generated distributions using the features extracted by the inception network (Szegedy et al., 2015). The lower FID indicates that the two distributions are more similar. To measure realism of the generated images, we also present human evaluation results using Amazon Mechanical Turk (AMT). Detailed evaluation protocols are described in Appendix D.1.

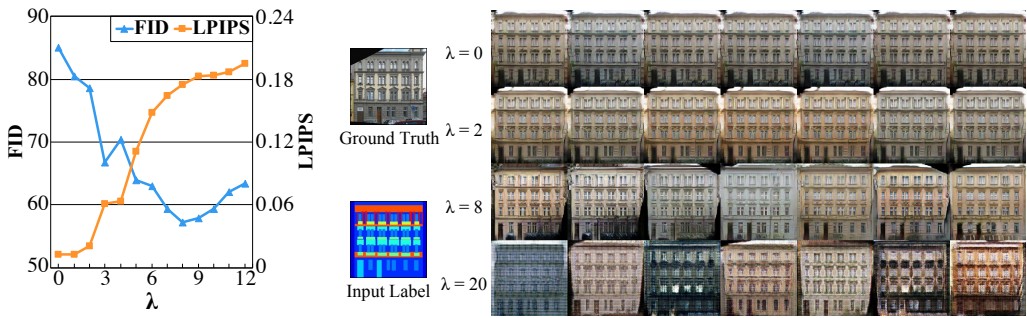

(a) LPIPS and FID scores      (b) Qualitative generation results with randomly sampled images

Figure 1: Impact of our regularization on multi-modal conditional generation.

---

[1] We use $l_1$ instead of $l_2$ norm for consistency with reconstruction loss (Isola et al., 2017). We also tested with $l_2$ and observe minor differences.

**Impact of the Proposed Regularization.** To analyze the impact of our regularization on learning a multi-modal mapping, we first conduct an ablation study by varying the weights ($\lambda$) for our regularization. We choose *label→image* dataset for this experiment, and summarize the results in Figure 1. From the figure, it is clearly observed that the baseline cGAN ($\lambda = 0$) experiences a severe mode-collapse and produces deterministic outputs. By adding our regularization ($\lambda > 0$), we observe that the diversity emerges from the generator outputs. Increasing the $\lambda$ increases LPIPS scores and lower the FID, which means that the generator learns a more diverse mapping from input to output, and the generated distribution is getting closer to the actual distribution. If we impose too strong constraints on diversity with high $\lambda$, the diversity keeps increasing, but generator outputs become less realistic and deviate from the actual distribution as shown in high FID (*i.e.* we got FID= 191 and LPIPS=0.20 for $\lambda = 20$). It shows that there is a natural trade-off between realism and diversity, and our method can control a balance between them by controlling $\lambda$.

**Comparison with BicycleGAN (Zhu et al., 2017b).** Next, we conduct comparison experiments with BicycleGAN (Zhu et al., 2017b), which is proposed to achieve multi-modal conditional generation in image-to-image translation. In this experiment, we fix $\lambda = 8$ for our method across all datasets and compare it against BicycleGAN with its optimal settings. Table 1 summarizes the results. Compared to the cGAN baseline, both our method and BicycleGAN are effective to learn multi-modal output distributions as shwon in higher LPIPS scores. Compared to BicycleGAN, our method still generates much diverse outputs and distributions that are generally more closer to actual ones as shown in lower FID score. In human evaluation on perceptual realism, we found that there is no clear winning method over others. It indicates that outputs from all three methods are in similar visual quality. Note that applying BicycleGAN to baseline cGAN requires non-trivial modifications in network architecture and obejctive function, while the proposed regularization can be simply integrated into the objective function without any modifications. Figure 2 illustrates generation results by our method. See Appendix D.1.3 for qualitative comparisons to BicycleGAN and cGAN.

We also conducted an experiment by varying a length of latent code $\boldsymbol{z}$. Table 2 summarizes the results. As discussed in Zhu et al. (2017b), generation quality of BicycleGAN degrades with high-dimensional $\boldsymbol{z}$ due to the difficulties in matching the encoder distribution $E(\boldsymbol{x})$ with prior distribution $p(\boldsymbol{z})$. Compared to BicycleGAN, our method is less suffered from such issue by sampling $\boldsymbol{z}$ from the prior distribution, thus exhibits consistent performance over various latent code sizes.

| Method | *label→image* | | *map→image* | | *edge→photo* | |
|---|---|---|---|---|---|---|
| | FID | LPIPS | FID | LPIPS | FID | LPIPS |
| cGAN | 85.07 | 0.01 | 90.08 | 0.02 | 31.80 | 0.02 |
| BicycleGAN | 62.95 | 0.15 | 55.53 | 0.11 | **20.27** | 0.11 |
| DSGAN | **57.20** | **0.18** | **49.92** | **0.13** | 23.06 | **0.12** |

Table 1: Comparisons of cGAN baseline, BicycleGAN and DSGAN (ours).

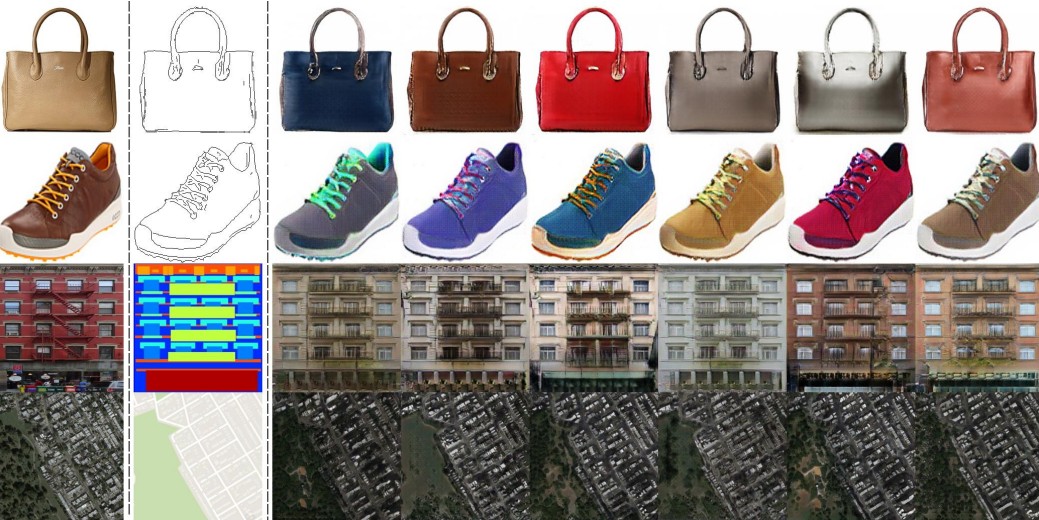

Figure 2: Diverse outputs generated by DSGAN. The first and second column shows ground-truth and input images, while the rest columns are generated images with different latent codes.

|  | $\|z\| = 8$ | | $\|z\| = 32$ | | $\|z\| = 64$ | | $\|z\| = 256$ | |
|  | FID | LPIPS | FID | LPIPS | FID | LPIPS | FID | LPIPS |
|---|---|---|---|---|---|---|---|---|
| BicycleGAN | 62.95 | 0.15 | 79.31 | 0.16 | 94.47 | 0.17 | 111.45 | 0.17 |
| DSGAN | **57.20** | **0.18** | **58.34** | **0.18** | **59.81** | **0.18** | **60.79** | **0.18** |

Table 2: Comparisons of BicycleGAN and DSGAN (ours) using various lengths of latent code.

|  | FID | LPIPS | Segmentation acc (%) |
|---|---|---|---|
| cGAN (pix2pixHD) | 48.85 | 0.00 | **0.93** |
| BicycleGAN | 89.42 | **0.16** | 0.72 |
| DSGAN (pix2pixHD) | **28.80** | 0.12 | 0.92 |

Table 3: Comparisons of high-resolution image synthesis results in Cityscape dataset.

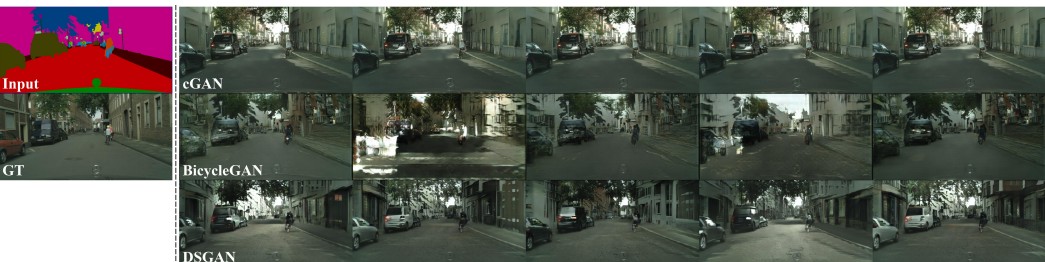

Figure 3: Qualitative comparison for high-resolution image synthesis ($1024 \times 512$ px.).

**Extension to High-Resolution Image Synthesis.** The proposed regularization is agnostic to the choice of network architecture and loss, therefore can be easily applicable to various methods. To demonstrate this idea, we apply our regularization to the network of pix2pixHD (Wang et al., 2018), which synthesizes a photo-realistic image of $1024 \times 512$ resolution from a segmentation label. In addition to the network architectures, Wang et al. (2018) incorporates a feature matching loss based on the discriminator as a reconstruction loss in Eq. (5). Therefore, this experiment also demonstrates that our regularization is compatible with other choices of $\mathcal{L}_{rec}$.

Table 3 shows the comparison results on Cityscape dataset (Cordts et al., 2016). In addition to FID and LPIPS scores, we compute the segmentation accuracy to measure the visual quality of the generated images. We compare the pixel-wise accuracy between input segmentation label and the predicted one from the generated image using DeepLab V3. (Chen et al., 2018). Since applying BicycleGAN to this baseline requires non-trivial modifications, we compared against the original BicycleGAN. As shown in the table, applying our method to the baseline effectively increases the output diversity with a cost of slight degradation in quality. Compared to BicycleGAN, our method generates much more visually plausible images. Figure 3 illustrates the qualitative comparison.

## 5.2 Image Inpainting

In this section, we demonstrate an application of our regularization to image inpainting task. The objective of this task is learning a generator $G : \mathcal{X} \to \mathcal{Y}$ that takes an image with missing regions $\boldsymbol{x} \in \mathcal{X}$ and generates a complete image $\boldsymbol{y} \in \mathcal{Y}$ by inferring the missing regions.

For this task, we employ generator and discriminator networks from Iizuka et al. (2017) as a baseline cGAN model with minor modification (See Appendix for more details). To create a data for inpainting, we take $256 \times 256$ images of centered faces from the celebA dataset (Liu et al., 2015) and remove center pixels of size $128 \times 128$ which contains most parts of the face. Similar to the image-to-image task, we employ FID and LPIPS to measure the generation performance. Please refer Appendix D.2 for more details about the network architecture and implementation details.

In this experiment, we also test with an extension of our regularization using a different sample distance metric. Instead of computing sample distances directly from the generator output as in Eq. (2), we use the *encoder* features that capture more semantically meaningful distance between samples. Similar to feature matching loss (Wang et al., 2018), we use the features from a discriminator to compute our regularization as follow:

$$\mathcal{L}_{\boldsymbol{z}}(G) = \mathbb{E}_{\boldsymbol{z}_1, \boldsymbol{z}_2} \left[ \frac{\frac{1}{L} \sum_{l=1}^{L} \left\| D^l(\boldsymbol{x}, \boldsymbol{z}_1) - D^l(\boldsymbol{x}, \boldsymbol{z}_2) \right\|}{\|\boldsymbol{z}_1 - \boldsymbol{z}_2\|} \right], \tag{6}$$

where $D^l$ indicates a feature extracted from $l_{\text{th}}$ layer of the discriminator $D$. We denote our methods based on Eq. (2) and Eq. (6) as $\text{DSGAN}_{\text{RGB}}$ and $\text{DSGAN}_{\text{FM}}$, respectively. Since there is no prior work on stochastic image inpainting to our best knowledge, we present comparisons of cGAN baseline along with our variants.

**Analysis on Regularization.** We conduct both quantitative and qualitative comparisons of our methods and summarize the results in Table 4 and Figure 4, respectively. As we observed in the previous section, adding our regularization induces multi-modal outputs from the baseline cGAN. See Figure F for qualitative impact of $\lambda$. Interestingly, we can see that sample variations in $\text{DSGAN}_{\text{RGB}}$ tend to be in a low-level (*e.g.* global skin-color). We believe that sample difference in color may not be appropriate for faces, since human reacts more sensitively to the changes in semantic features (*e.g.* facial landmarks) than just color. Employing perceptual distance metric in our regularization leads to semantically more meaningful variations, such as expressions, identity, *etc.*.

|  | FID | LPIPS |
|---|---|---|
| cGAN | 13.99 | 0.00 |
| $\text{DSGAN}_{\text{RGB}}$ | 13.95 | 0.01 |
| $\text{DSGAN}_{\text{FM}}$ | **13.94** | **0.05** |

Table 4: Quantitative comparisons of our variants.

Figure 4: Qualitative comparisons of our variants. We present one example for baseline as it produces deterministic outputs.

**Analysis on Latent Space.** To further understand if our regularization encourages $z$ to encode meaningful features, we conduct qualitative analysis on $z$. We employ $\text{DSGAN}_{\text{FM}}$ for this experiments. We generate multiple samples across various input conditions while fixing the latent codes $z$. Figure 5 illustrates the results. We observe that our method generates outputs which are realistic and diverse depending on $z$. More interestingly, the generator outputs given the same $z$ exhibit similar attributes (*e.g.* gaze direction, smile) but also context-specific characteristics that match the input condition (*e.g.* skin color, hairs). It shows that our method guides the generator to learn meaningful latent factors in $z$, which are disentangled from the input context to some extent.

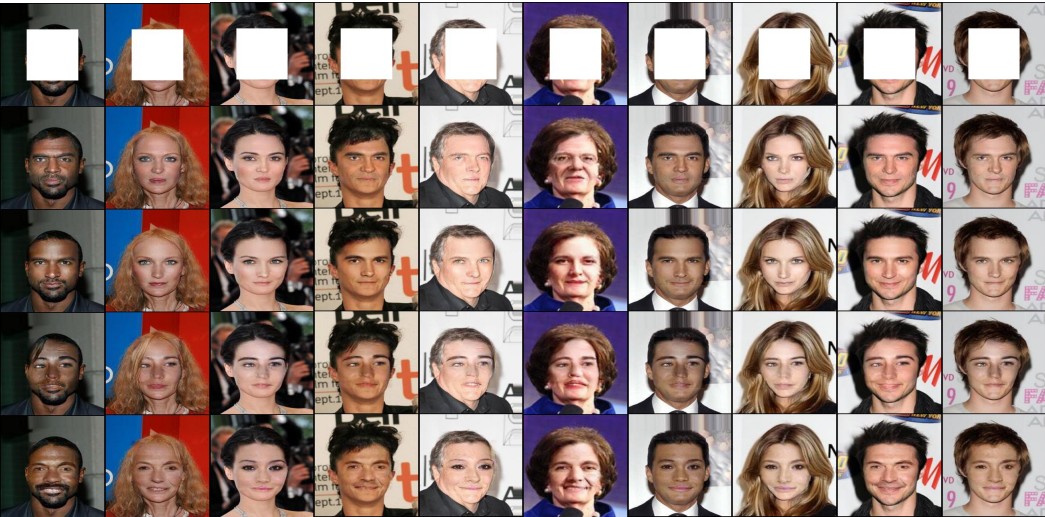

Figure 5: Stochastic image inpainting results. Given an input image with missing region (first row), we generate multiple faces by sampling different $z$ (second–fifth rows). Each row is generated from the same $z$, and exhibits similar face attributes.

## 5.3 VIDEO PREDICTION

In this section, we apply our method to a conditional *sequence* generation task. Specifically, we consider a task of anticipating $T$ future frames $\{x_{K+1}, x_{K+2}, ..., x_{K+T}\} \in \mathcal{Y}$ conditioned on $K$ previous frames $\{x_1, x_2, ..., x_K\} \in \mathcal{X}$. Since both the input and output of the generator are sequences in this task, we simply modify our regularization by

$$\mathcal{L}_z(G) = \mathbb{E}_{z_1, z_2} \left[ \frac{\frac{1}{T} \sum_{t=K}^{K+T} \|G(x_{1:t}, z_1) - G(x_{1:t}, z_2)\|_1}{\|z_1 - z_2\|_1} \right], \tag{7}$$

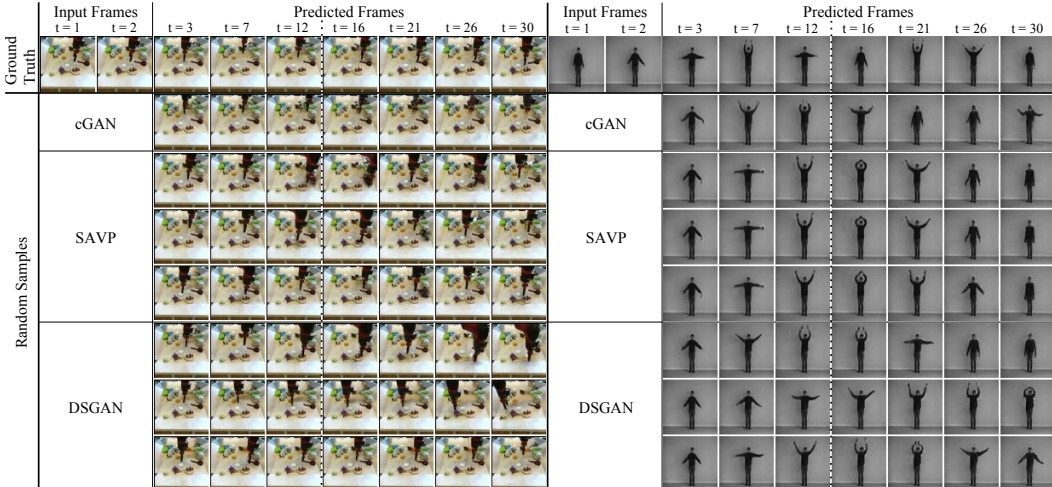

Figure 6: Stochastic video prediction results. Given two input frames, we present three random samples generated by each method. Compared to the baseline that produces deterministic outputs and SAVP that has limited diversity in KTH, our method generates diverse futures in both datasets.

(Base: $1.0 \times 10^{-3}$)

| | BAIR | | | KTH | | |
|---|---|---|---|---|---|---|
| Method | Diversity | $Sim_{max}$ | $Dist_{min}$ | Diversity | $Sim_{max}$ | $Dist_{min}$ |
| cGAN | 2.48 | 861.92 | 22.46 | 0.04 | 802.79 | 5.00 |
| SAVP | 18.93 | 869.58 | 20.44 | 0.51 | 777.70 | 5.48 |
| DSGAN | **26.75** | **874.12** | **18.46** | **3.96** | **855.10** | **3.84** |

Table 5: Comparisons of cGAN, SAVP, and DSGAN (ours). *Diversity*: pixel-wise distance among the predicted videos. $Sim_{max}$: largest cosine similarity between the predicted video and the ground truth. $Dist_{min}$: closest pixel-wise distance between the predicted video and the ground truth.

where $x_{1:t}$ represents a set of frames from time step 1 to $t$.

We compare our method against SAVP (Lee et al., 2018), which also addresses the multi-modal video prediction task. Similar to Zhu et al. (2017b), it employs a hybrid model of conditional GAN and VAE, but using the recurrent generator designed specifically for future frame prediction. We take only GAN component (generator and discriminator networks) from SAVP as a baseline cGAN model and apply our regularization with $\lambda = 50$ to induce stochasticity. We use $|z| = 8$ for all compared methods. See Appendix D.3.2 for more details about the network architecture.

We conduct experiments on two datasets from the previous literature: the BAIR action-free robot pushing dataset (Ebert et al., 2017) and the KTH human actions dataset (Schuldt et al., 2004). To measure both the diversity and the quality, we generate 100 random samples of 28 future frames for each test video and compute the (a) *Diversity* (pixel-wise distance among the predicted videos) and the (b) $Dist_{min}$ (minimum pixel-wise distance between the predicted videos and the ground truth). Also, for a better understanding of quality, we additionally measured the (c) $Sim_{max}$ (largest cosine similarity metric between the predicted video and the ground truth on VGGNet (Simonyan & Zisserman, 2015) feature space). An ideal stochastic video prediction model may have higher *Diversity*, while having lower $Dist_{min}$ with higher $Sim_{max}$ so that a model still can predict similar to the ground truth as a candidate. More details about evaluation metric are described in Appendix D.3.3.

We present both quantitative and qualitative comparison results in Table 5 and Figure 6, respectively. As illustrated in the results, both our method and SAVP can predict diverse futures compared to the baseline cGAN that produces deterministic outputs. As shown in Table 5, our method generates more diverse and realistic outputs than SAVP with much less number of parameters and simpler training procedures. Interestingly, as shown in KTH results, SAVP still suffers from a mode-collapse problem when the training videos have limited diversity, whereas our method generally works well in both cases. It shows that our method generalizes much better to various videos despite its simplicity.

## 6 CONCLUSION

In this paper, we investigate a way to resolve a mode-collapsing in conditional GAN by regularizing generator. The proposed regularization is simple, general, and can be easily integrated into existing

conditional GANs with broad classes of loss function, network architecture, and data modality. We apply our regularization for three conditional generation tasks and show that simple addition of our regularization to existing cGAN objective effectively induces the diversity. We believe that achieving an appropriate balance between realism and diversity by *learning* $\lambda$ and $\tau$ such that the learned distribution matches an actual data distribution would be an interesting future work.

**Acknowledgement**    This work was supported in part by ONR N00014-13-1-0762, NSF CAREER IIS-1453651, DARPA Explainable AI (XAI) program #313498, and Sloan Research Fellowship.

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

APPENDIX

## A  DERIVATION OF LOWER-BOUND OF GRADIENT NORM

In this section, we provide a derivation of our regularization term from a true gradient norm of the generator. Given arbitrary latent samples $z_1, z_2$, from gradient theorem we have

$$
\begin{aligned}
\frac{\|G(\boldsymbol{x}, \boldsymbol{z}_2) - G(\boldsymbol{x}, \boldsymbol{z}_1)\|}{\|\boldsymbol{z}_2 - \boldsymbol{z}_1\|} &= \frac{\|\int_{\gamma[\boldsymbol{z}_1, \boldsymbol{z}_2]} \nabla_{\boldsymbol{z}} G(\boldsymbol{x}, \boldsymbol{z}) \cdot d\boldsymbol{z}\|}{\|\boldsymbol{z}_2 - \boldsymbol{z}_1\|} \\
&= \frac{\|\int_0^1 \nabla_{\boldsymbol{z}} G(\boldsymbol{x}, \boldsymbol{\gamma}(t)) \cdot \boldsymbol{\gamma}'(t) dt\|}{\|\boldsymbol{z}_2 - \boldsymbol{z}_1\|} \\
&= \frac{\|\int_0^1 \nabla_{\boldsymbol{z}} G(\boldsymbol{x}, \boldsymbol{\gamma}(t)) \cdot (\boldsymbol{z}_2 - \boldsymbol{z}_1) dt\|}{\|\boldsymbol{z}_2 - \boldsymbol{z}_1\|} \\
&\leq \frac{\int_0^1 \|\nabla_{\boldsymbol{z}} G(\boldsymbol{x}, \boldsymbol{\gamma}(t))\| \|\boldsymbol{z}_2 - \boldsymbol{z}_1\| dt}{\|\boldsymbol{z}_2 - \boldsymbol{z}_1\|} \\
&= \int_0^1 \|\nabla_{\boldsymbol{z}} G(\boldsymbol{x}, \boldsymbol{\gamma}(t))\| dt \;,
\end{aligned}
\tag{8}
$$

where $\gamma$ is a straight line connecting $z_1$ and $z_2$, where $\gamma(0) = z_1$ and $\gamma(1) = z_2$.

Apply expectation on both sides of (8) with respect to $z_1, z_2$ from standard Gaussian distribution gives Eqn. 4:

$$
\mathbb{E}_{\boldsymbol{z}_1, \boldsymbol{z}_2} \left[ \frac{\|G(\boldsymbol{x}, \boldsymbol{z}_2) - G(\boldsymbol{x}, \boldsymbol{z}_1)\|}{\|\boldsymbol{z}_2 - \boldsymbol{z}_1\|} \right] \leq \mathbb{E}_{\boldsymbol{z}_1, \boldsymbol{z}_2} \left[ \int_0^1 \|\nabla_{\boldsymbol{z}} G(\boldsymbol{x}, \boldsymbol{\gamma}(t))\| dt \right] \;.
\tag{9}
$$

## B  COLLAPSING TO A MODE AS A GROUP

For notational simplicity, we omit the conditioning variable from the generator and focus on a mapping of latent code to output $G_\theta : \mathcal{Z} \to \mathcal{Y}$ where $\mathcal{Y}$ is the image space.

**Definition B.1.** A mode $\mathcal{M}$ is a subset of $\mathcal{Y}$ satisfying $\max_{\boldsymbol{y} \in \mathcal{M}} \|\boldsymbol{y} - \boldsymbol{y}^*\| < \alpha$ for some image $\boldsymbol{y}^*$ and $\alpha > 0$. Let $z_1$ be a sample in latent space, we say $z_1$ is attracted to a mode $\mathcal{M}$ by $\epsilon$ from a gradient step if $\|\boldsymbol{y}^* - G_{\theta_{t+1}}(\boldsymbol{z}_1)\| + \epsilon < \|\boldsymbol{y}^* - G_{\theta_t}(\boldsymbol{z}_1)\|$, where $\boldsymbol{y}^* \in \mathcal{M}$ is an image in a mode, $\theta_t$ and $\theta_{t+1}$ are the generator parameters before and after the gradient updates respectively.

In other words, we define modes as sets consisting of images that are close to some real images, and we consider a situation where the generator output $G_{\theta_t}(\boldsymbol{z}_1)$ at certain $z_1$ is attracted to a mode $\mathcal{M}$ by a single gradient update.

With Definition B.1, we are now ready to state and prove the following proposition.

**Proposition B.1.** Suppose $z_1$ is attracted to the mode $\mathcal{M}$ by $\epsilon$, then there exists a neighborhood $\mathcal{N}_r(\boldsymbol{z}_1)$ of $z_1$ such that $z_2$ is attracted to $\mathcal{M}$ by $\epsilon/2$, for all $\boldsymbol{z}_2 \in \mathcal{N}_r(\boldsymbol{z}_1)$. The size of $\mathcal{N}_r(\boldsymbol{z}_1)$ can be arbitrarily large but is bounded by an open ball of radius $r$ where

$$
r = \epsilon \cdot \left( 4 \inf_{\boldsymbol{z}} \left\{ \max \left( \frac{\|G_{\theta_t}(\boldsymbol{z}_1) - G_{\theta_t}(\boldsymbol{z})\|}{\|\boldsymbol{z}_1 - \boldsymbol{z}\|}, \frac{\|G_{\theta_{t+1}}(\boldsymbol{z}_1) - G_{\theta_{t+1}}(\boldsymbol{z})\|}{\|\boldsymbol{z}_1 - \boldsymbol{z}\|} \right) \right\} \right)^{-1} \;.
\tag{10}
$$

*Proof.* Consider the following expansion.

$$
\begin{aligned}
\|\boldsymbol{y}^* - G_{\theta_{t+1}}(\boldsymbol{z}_2)\| &\leq \|\boldsymbol{y}^* - G_{\theta_{t+1}}(\boldsymbol{z}_1)\| + \|G_{\theta_{t+1}}(\boldsymbol{z}_1) - G_{\theta_{t+1}}(\boldsymbol{z}_2)\| \\
&< \|\boldsymbol{y}^* - G_{\theta_t}(\boldsymbol{z}_1)\| + \|G_{\theta_{t+1}}(\boldsymbol{z}_1) - G_{\theta_{t+1}}(\boldsymbol{z}_2)\| - \epsilon \qquad \text{(Definition B.1)} \\
&\leq \|\boldsymbol{y}^* - G_{\theta_t}(\boldsymbol{z}_2)\| + \|G_{\theta_t}(\boldsymbol{z}_2) - G_{\theta_t}(\boldsymbol{z}_1)\| + \|G_{\theta_{t+1}}(\boldsymbol{z}_1) - G_{\theta_{t+1}}(\boldsymbol{z}_2)\| - \epsilon \\
&= \|\boldsymbol{y}^* - G_{\theta_t}(\boldsymbol{z}_2)\| \\
&\quad + \left( \frac{\|G_{\theta_t}(\boldsymbol{z}_1) - G_{\theta_t}(\boldsymbol{z}_2)\|}{\|\boldsymbol{z}_1 - \boldsymbol{z}_2\|} + \frac{\|G_{\theta_{t+1}}(\boldsymbol{z}_1) - G_{\theta_{t+1}}(\boldsymbol{z}_2)\|}{\|\boldsymbol{z}_1 - \boldsymbol{z}_2\|} \right) \|\boldsymbol{z}_1 - \boldsymbol{z}_2\| - \epsilon.
\end{aligned}
\tag{11}
$$

Eq. (11) implies that

$$\|\boldsymbol{y}^* - G_{\theta_{t+1}}(\boldsymbol{z}_2)\| + \frac{\epsilon}{2} < \|\boldsymbol{y}^* - G_{\theta_t}(\boldsymbol{z}_2)\|, \tag{12}$$

for all $\boldsymbol{z}_2$ that satisfies

$$\left( \frac{\|G_{\theta_t}(\boldsymbol{z}_1) - G_{\theta_t}(\boldsymbol{z}_2)\|}{\|\boldsymbol{z}_1 - \boldsymbol{z}_2\|} + \frac{\|G_{\theta_{t+1}}(\boldsymbol{z}_1) - G_{\theta_{t+1}}(\boldsymbol{z}_2)\|}{\|\boldsymbol{z}_1 - \boldsymbol{z}_2\|} \right) \|\boldsymbol{z}_1 - \boldsymbol{z}_2\| \leq \frac{\epsilon}{2}. \tag{13}$$

Define $\mathcal{N}_\tau(\boldsymbol{z}_1) = \left\{ \boldsymbol{z} : \max \left( \frac{\|G_{\theta_t}(\boldsymbol{z}_1) - G_{\theta_t}(\boldsymbol{z})\|}{\|\boldsymbol{z}_1 - \boldsymbol{z}\|}, \frac{\|G_{\theta_{t+1}}(\boldsymbol{z}_1) - G_{\theta_{t+1}}(\boldsymbol{z})\|}{\|\boldsymbol{z}_1 - \boldsymbol{z}\|} \right) \leq \tau \right\}$, then Eq. (13) holds for any $\boldsymbol{z}_2 \in \bigcup_{\tau > 0} \left\{ B_{\epsilon/4\tau}(\boldsymbol{z}_1) \cap \mathcal{N}_\tau(\boldsymbol{z}_1) \right\} = \mathcal{N}_r(\boldsymbol{z}_1)$. We have $\mathcal{N}_r(\boldsymbol{z}_1) \neq \emptyset$ since $\boldsymbol{z}_1 \in \mathcal{N}_r(\boldsymbol{z}_1)$.

The size of $\mathcal{N}_r(\boldsymbol{z}_1)$ can be arbitrarily large if $B_{\epsilon/4\tau}(\boldsymbol{z}_1) \subseteq \mathcal{N}_\tau(\boldsymbol{z}_1)$ for arbitrarily small $\tau$. And for any $\tau > 0$ such that $\tau \leq \max \left( \frac{\|G_{\theta_t}(\boldsymbol{z}_1) - G_{\theta_t}(\boldsymbol{z})\|}{\|\boldsymbol{z}_1 - \boldsymbol{z}\|}, \frac{\|G_{\theta_{t+1}}(\boldsymbol{z}_1) - G_{\theta_{t+1}}(\boldsymbol{z})\|}{\|\boldsymbol{z}_1 - \boldsymbol{z}\|} \right)$ for all $\boldsymbol{z}$, we have $\mathcal{N}_r(\boldsymbol{z}_1) \subseteq B_{\epsilon/4\tau}(\boldsymbol{z}_1)$. In particular, pick the largest $\tau$ possible yields the bound $\mathcal{N}_r(\boldsymbol{z}_1) \subseteq B_r(\boldsymbol{z}_1)$,

$$r = \epsilon \cdot \left( 4 \inf_{\boldsymbol{z}} \left\{ \max \left( \frac{\|G_{\theta_t}(\boldsymbol{z}_1) - G_{\theta_t}(\boldsymbol{z})\|}{\|\boldsymbol{z}_1 - \boldsymbol{z}\|}, \frac{\|G_{\theta_{t+1}}(\boldsymbol{z}_1) - G_{\theta_{t+1}}(\boldsymbol{z})\|}{\|\boldsymbol{z}_1 - \boldsymbol{z}\|} \right) \right\} \right)^{-1}. \tag{14}$$

$\square$

## C  ABLATION STUDY

### C.1  APPLICATION TO UNCONDITIONAL GAN

Since the proposed regularization is not only limited to conditional GAN, we further analyze its impact on unconditional GAN. To this end, we adopt synthetic datasets from (Srivastava et al., 2017; Metz et al., 2017), a mixture of eight 2D Gaussian distributions arranged in a ring. For unconditional generator and discriminator, we adopt the vanilla GAN implementation from (Srivastava et al., 2017), and train the model with and without our regularization ($\lambda = 0.1$). We follow the same evaluation protocols used in (Srivastava et al., 2017). It generates 2,500 samples by the generator, and counts a sample as high-quality if it is within three standard deviations of the nearest mode. Then the performance is reported by 1) counting the number of modes containing at least one high-quality sample and 2) computing the portion of high-quality samples from all generated ones. We summarize the qualitative and quantitative results (10-run average) in Figure A and Table A, respectively.

As illustrated in Figure A, we observe that vanilla GAN experiences a severe mode collapse, which puts a significant probability mass around a single output mode. Contrary to results reported in (Srivastava et al., 2017), we observed that the mode captured by the generator is still not close enough to the actual mode, resulted in 0 high-quality samples as shown in Table A. On the other hand, applying our regularization effectively alleviates the mode-collapse problem by encouraging the generator to efficiently explore the data space, enabling the generator to capture much more modes compared to vanilla GAN setting.

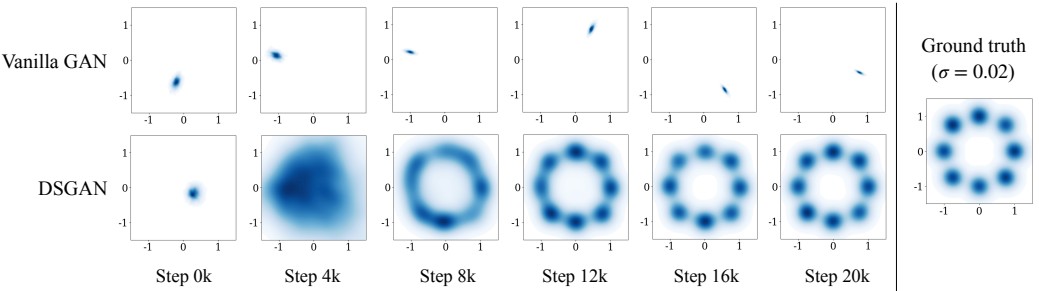

Figure A: Density plots of ground-truth, vanilla GAN and DSGAN.

|  | 2D Ring | |
|---|---|---|
| Method | Modes (Max 8) | High Quality Samples (%) |
| Vanilla GAN | 0 (1[*]) | 0.0 (99.3[*]) |
| Unrolled GAN | 7.6 | 35.6 |
| VEEGAN | 8 | 52.9 |
| DSGAN | 8 | **81.4** |

Table A: Quantitative evaluation results (* denotes the results reported in Srivastava et al. (2017)).

### C.2  IMPACT ON UNBALANCED cGAN

In principle, our regularization can help the generator to cope with vanishing gradient from the discriminator to some extent, as it spreads out the generator landscape thus increases the chance to capture useful gradient signals around true data points. To verify its impact, we simulate the vanishing gradient problem by training the baseline cGAN until it converges and retraining the model with our regularization while initializing the discriminator with the pre-trained weights. Empirically we observed that the pre-trained discriminator can distinguish the real data and generated samples from the randomly initialized generator almost perfectly, and the generator experiences a severe vanishing gradient problem at the beginning of the training. We use the image-to-image translation model on *label→image* dataset for this experiment.

In the experiment, we found that the generator converges to the FID and LPIPS scores of 52.32 and 0.16, respectively, which are close to the ones we achieved with the balanced discriminator (FID: 57.20, LPIPS: 0.18). We observed that our regularization loss goes down very quickly in the

early stage of training, which helps the generator to efficiently explorer its output space when the discriminator gradients are vanishing. Together with the reconstruction loss, we found that it helps the generator to capture useful learning signals from the discriminator and learn both realistic and diverse modes in the conditional distribution.

# D ADDITIONAL EXPERIMENT RESULTS

This section provides additional experiment details and results that could not be accommodated in the main paper due to space restriction. We are going to release the code and datasets upon the acceptance of the paper.

## D.1 IMAGE-TO-IMAGE TRANSLATION

### D.1.1 BASELINE MODELS

**BicycleGAN's Generator and Discriminator.** We use BicycleGAN's generator and discriminator structures as a baseline cGAN. The baseline model has exactly the same hyperparameters as BicycleGAN including the weight of pixel-wise L1 loss and GAN loss. The baseline model setting then basically becomes a pix2pix image-to-image translation setting (Isola et al., 2017). The generator architecture is a U-Net style network (Ronneberger et al., 2015) and the discriminator is a two-scale patchGAN-style (Li & Wand, 2016) network.

**Pix2pixHD Baseline.** In this setting, we adopt the generator and discriminator networks from pix2pixHD (Wang et al., 2018) as a baseline cGAN. Compared to the original pix2pixHD network that employs two nested generators, we use only one generator for simplicity. Since the original generator does not contain stochastic component, we modified the generator network by injecting the latent code after the downsampling layers by spatial tiling and depth-wise concatenation. The discriminator is a two-scale patchGAN-style (Li & Wand, 2016) network. Following the original setting, we employ feature matching loss based on discriminator (Wang et al., 2018) and perceptual loss based on the pre-trained VGGNet (Simonyan & Zisserman, 2015) as a reconstruction loss in Eq. (5).

### D.1.2 EVALUATION METRICS

Here we provide a detailed descriptions for evaluation metrics and evaluation protocols.

**Learned Perceptual Image Patch Similarity, LPIPS (Zhang et al., 2018).** LPIPS score measures the diversity of the generated samples using the L1 distance of features extracted from pre-trained AlexNet (Krizhevsky et al., 2012). We generate 20 samples for each validation image, and compute the average of pairwise distances between all samples generated from the same input. Then we report the average of LPIPS scores over all validation images.

**Fréchet Inception Distance, FID (Heusel et al., 2017).** For each method, we compute FID score on the validation dataset. For each input from the validation dataset, we sample 20 randomly generated output. We take the generated images as a generated dataset and compute the FID score between the generated dataset and training dataset. If the size of an image is different between the training dataset and generated dataset, we resize training images to the size of generated images. We use the features from the final average pooling layer of the InceptionV3 (Szegedy et al., 2015) network to compute Fréchet Distance.

**Human Evaluation via Amazon Mechanical Turk (AMT).** To compare the perceptual quality of generations among different methods, we conduct human evaluation via AMT. We conduct side-by-side comparisons between our method and a competitor (*i.e.* baseline cGAN and BicycleGAN). Specifically, we present two sets of images generated by each compared method given the same input condition, and ask turkers to choose the set that is visually more plausible and matches the input condition. Each set has 6 randomly sampled images. We collect answers over 100 examples for each dataset, where each question is answered by 5 unique turkers.

### D.1.3 QUALITATIVE RESULTS

**Qualitative Comparison.** We present the qualitative comparisons of various methods presented in Table 1 and Table 3 in the main paper. Figure B illustrates the qualitative comparison results of DS-GAN (ours), baseline cGAN and BicycleGAN in Table 1. In the example of *edges→photo* dataset,

the input edge images miss some of the features in the ground-truth images (*e.g.* shoelace). While both DSGAN and baseline cGAN are able to capture such missing parts in an input, we observe that some of BicycleGAN's outputs are missing it. Also in the example of *maps→images* dataset, both DSGAN and baseline cGAN can generate natural and variable vegetation in the corresponding area while BicycleGAN tends to generate plain texture with global color variations in such area. These two examples show how our regularization can help cGAN to learn more visually reasonable and diverse results. We additionally present the qualitative comparison results of Table 3. We observe that the generation results from BicycleGAN suffers from low-visual quality, while our method is able to generate fine details of the objects and scene by exploiting the network for high-resolution image synthesis.

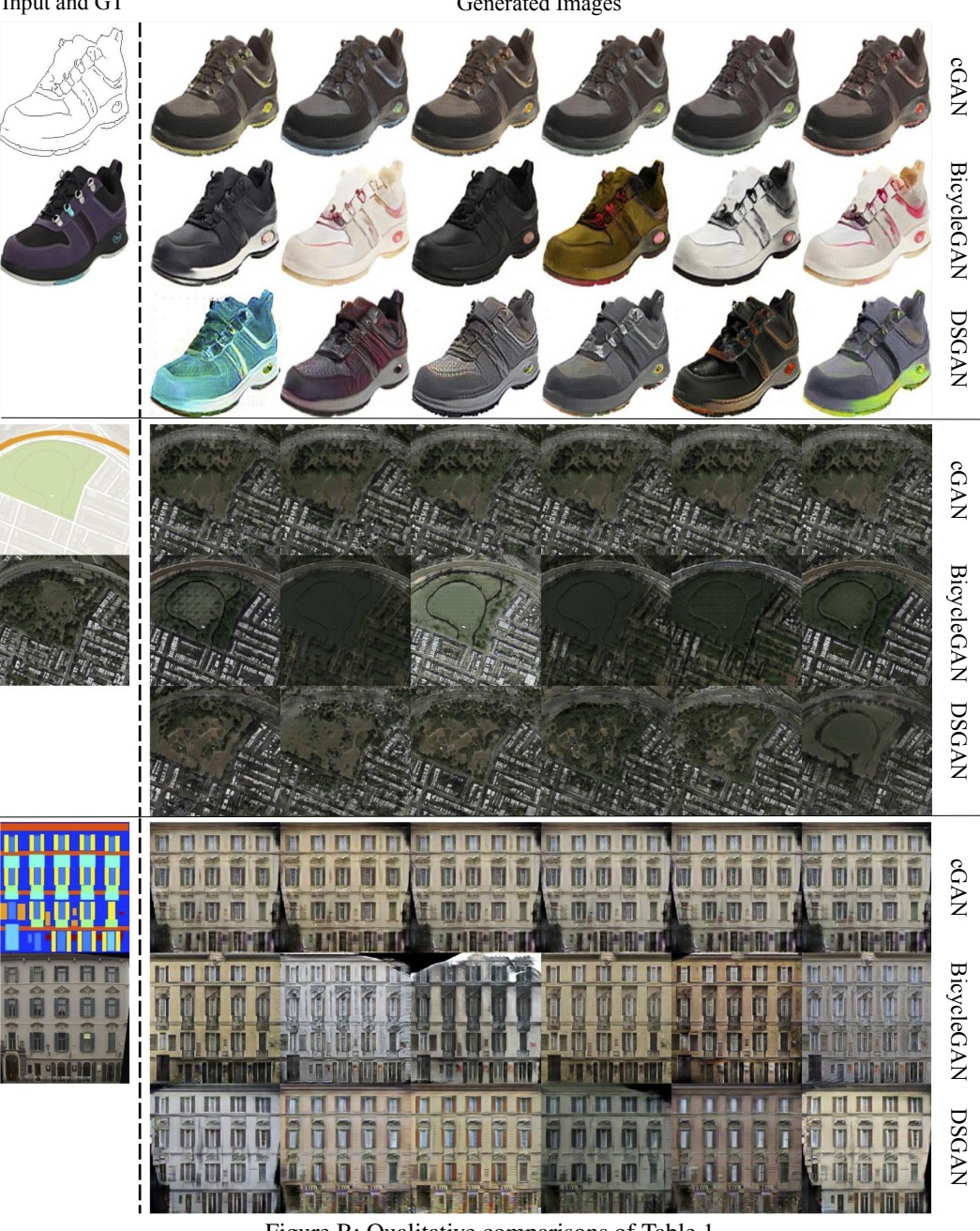

Figure B: Qualitative comparisons of Table 1.

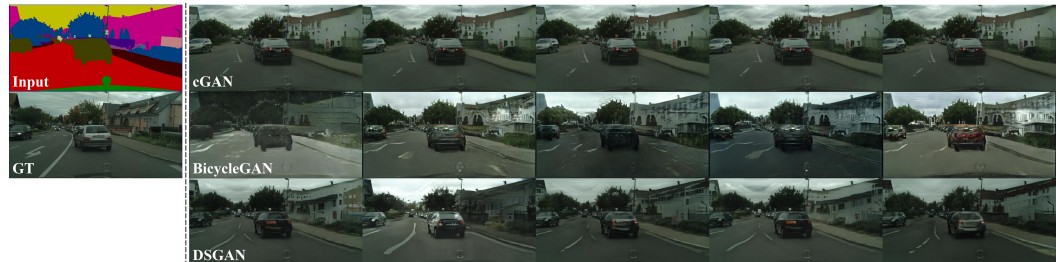

Figure C: Qualitative comparisons of Table 3.

**Analysis on Latent Space.**   To better understand the latent space learned with the proposed regularization, we generate images in Cityscape dataset by interpolating two randomly sampled latent vectors by spherical linear interpolation (White, 2016). Figure D illustrates the interpolation results. As shown in the figure, the intermediate generation results are all reasonable and exhibit smooth transitions, which implies that the learned latent space has a smooth manifold.

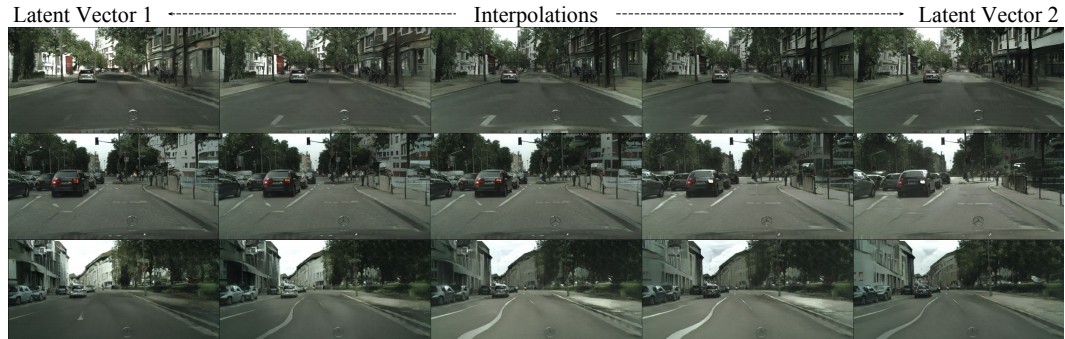

Figure D: Interpolation in latent space of DSGAN on Cityscapes dataset.

We also present the comparison of interpolation results between DSGAN and BicycleGAN on *maps → images* dataset. As shown in Figure E, DSGAN generates meaningful and diverse predictions on ambiguous regions (*e.g.* forest on a map) and has a smooth transition from one latent code to another. On contrary, the BicyceGAN does not show meaningful changes within the interpolations and sometimes has a sudden changes on its output (*e.g.* last generated image). We also observe similar patterns across many examples in this dataset. It shows an example that DSGAN learns better latent space than BicycleGAN.

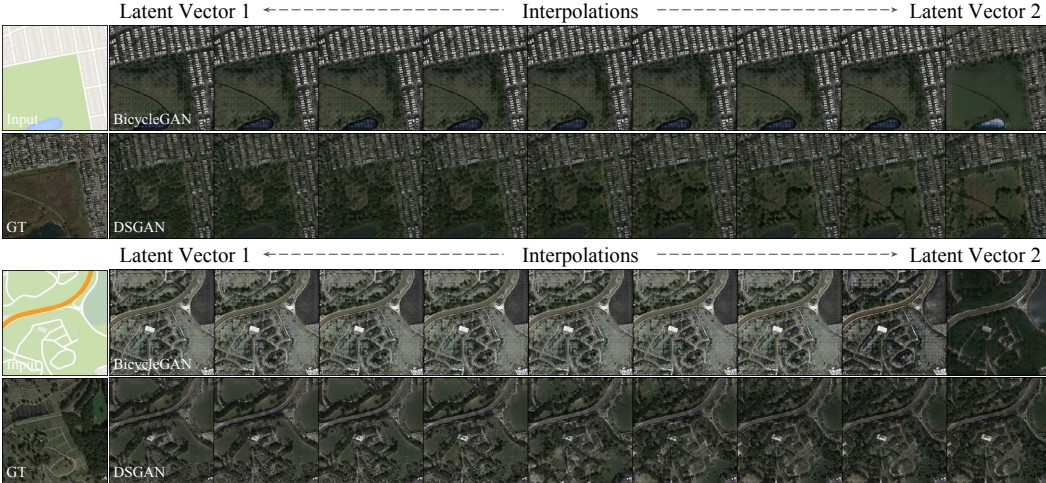

Figure E: Comparison of latent space interpolation results between DSGAN and BicycleGAN.

## D.2 IMAGE INPAINTING

In this section, we provide details of image inpainting experiment.

**Network Architecture.**   We employ the generator and discriminator networks from Iizuka et al. (2017) as baseline conditional GAN. Our generator takes $256 \times 256$ image with the masked region $\boldsymbol{x}$ as an input and produces $256 \times 256$ prediction of the missing region $\hat{\boldsymbol{y}}$ as an output. Then we combine the predicted image with the input by $\boldsymbol{y} = (1 - M) \odot \boldsymbol{x} + M \odot \hat{\boldsymbol{y}}$ as an output of the network, where $M$ is a binary mask indicating the missing region. Then the combined output $\boldsymbol{y}$ is passed as an input to the discriminator. We apply two modifications to the baseline model to achieve better generation quality. First, compared to the original model that employs the Mean Squared Error (MSE) as a reconstruction loss $\mathcal{L}_{rec}(G)$ in Eq. (5), we apply the feature matching loss based on the discriminator (Wang et al., 2018). Second, compared to the original model that employs two discriminators applied independently to the inpainted region and entire image, we employ only one discriminator on the inpainted region but using patchGAN-style discriminator (Li & Wand, 2016). Please note that these modifications are to achieve better image quality but irrelevant to our regularization.

**Analysis on Regularization.**   First, we conduct qualitative analysis on how the proposed regularization controls a diversity of the generator outputs. To this end, we train the model (DSGAN$_{\text{FM}}$) by varying the weights for our regularization, and present the results in Figure F. As already observed in Section 5.1, imposing stronger constraints on the generator by our regularization indeed increases the diversity in the generator outputs. With small weights (*e.g.* $\lambda = 2$), we observe limited visual differences among samples, such as subtle changes in facial expressions or makeup. By increasing $\lambda$ (*e.g.* $\lambda = 5$), we can see that more meaningful diversity emerges such as hair-style, age, and even identity while maintaining the visual quality and alignment to input condition. It shows more intuitively how our regularization can effectively help the model to discover more meaningful modes in the output space.

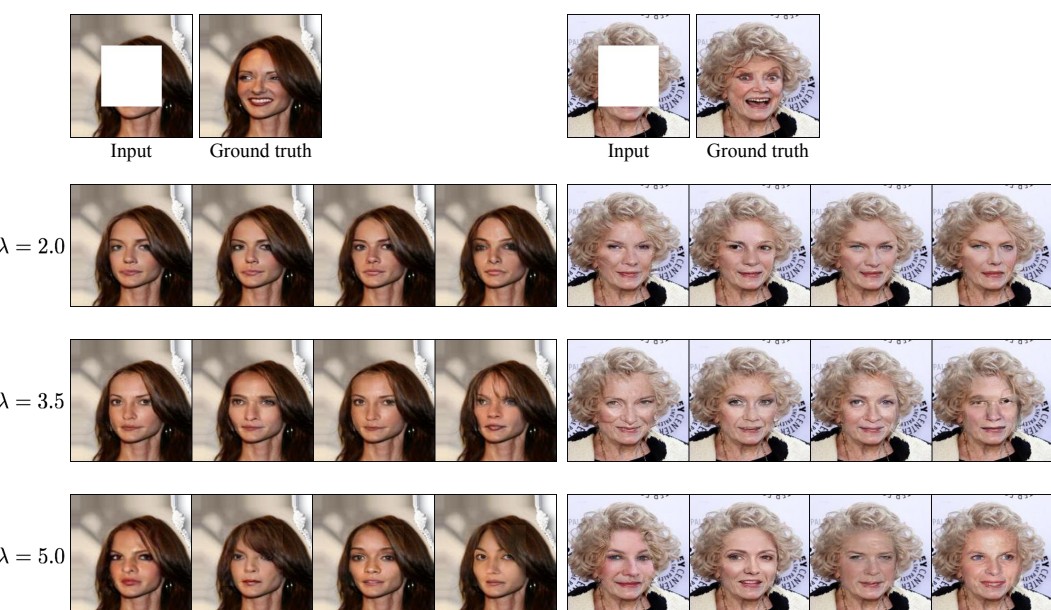

Figure F: Image inpainting results with different $\lambda$. We observe more diversity emerges from the genrator outputs as we increase the weights for our regularization.

**Analysis on Latent Space.**   We further conduct a qualitative analysis on the learned latent space. To verify that the model learns a continuous conditional distribution with our regularization, we conduct the interpolation experiment similar to the previous section. Specifically, we sample two random latent codes from the prior distribution and generate images by linearly interpolating the

latent code between two samples. Figure G illustrates the results. As it shows, the generator outputs exhibit a smooth transition between two samples, while most intermediate samples also look realistic.

Latent vector 1 --------------------------------- Interpolations --------------------------------- Latent vector 2

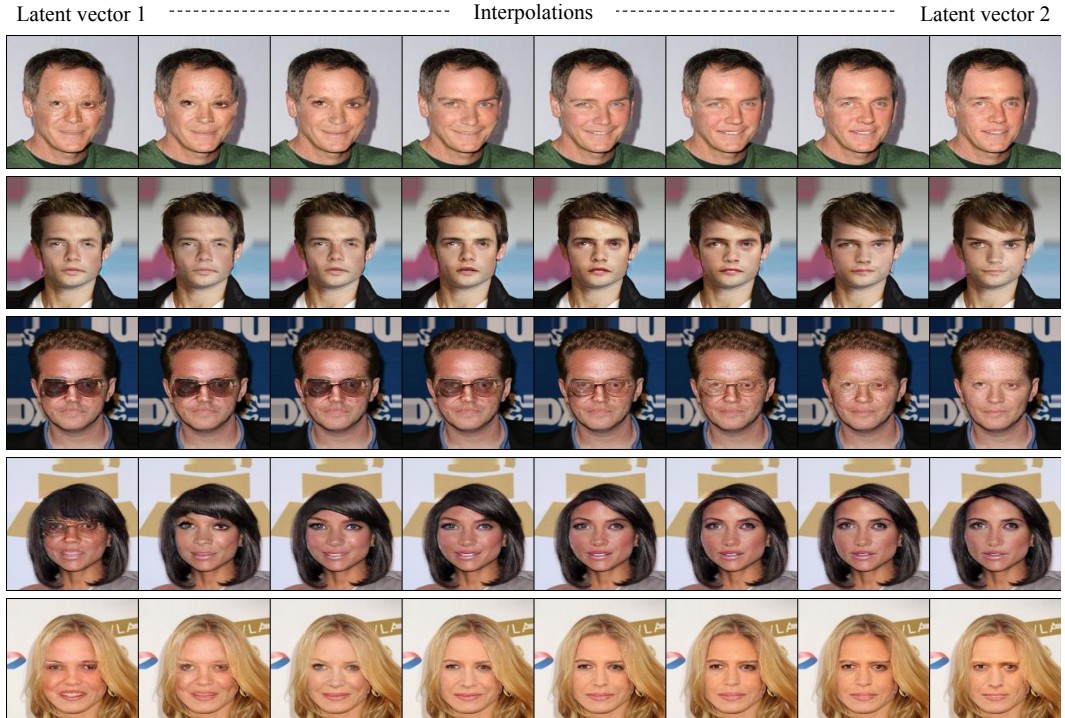

Figure G: Interpolation results on image inpainting task. For each row, we sample the two latent codes (leftmost and rightmost images), and generate the images from the interpolated latent codes from one latent code to another.

### D.3 VIDEO PREDICTION

In this section, we provide more details on network architecture, datasets and evaluation metrics on the video prediction task.

#### D.3.1 DATASET

We measure the effectiveness of our method based on two real-world datasets: the BAIR action-free robot pushing dataset (Ebert et al., 2017) and the KTH human actions dataset (Schuldt et al., 2004). For both of the dataset, we provide two frames as the condition and train the model to predict 10 future frames ($k = 2$, $T = 10$ in Eq. (7)). In testing time, we run each model to predict 28 frames ($k = 2$, $T = 28$). Following Lee et al. (2018), we used $64 \times 64$ frames for both datasets. The details of data pre-processing are described in below.

**BAIR Action-Free (Ebert et al., 2017).**   This dataset contains randomly moving robot arms on a table with a static background. This dataset contains the diverse movement of a robot arm with a diverse set of objects. We downloaded the pre-processed data provided by the authors (Lee et al., 2018) and used it directly for our experiment.

**KTH (Schuldt et al., 2004).**   Each video in this dataset contains a single person in a static background performing one of six activities: walking, jogging, running, boxing, hand waving, and hand clapping. We download the pre-processed videos from Villegas et al. (2017); Denton & Fergus (2018), which contains the frames with reasonable motions. Following Jang et al. (2018), we added a diversity to videos by randomly skipping frames in a range of [1,3].

#### D.3.2 NETWORK ARCHITECTURE

We compare our method against SAVP (Lee et al., 2018) which is proposed to achieve stochastic video prediction. SAVP addresses a mode-collapse problem using the hybrid model of conditional GAN and VAE. For a fair comparison, we construct our baseline cGAN by taking GAN component from SAVP (the generator and discriminator networks). In below, we provide more details of the generator and discriminator architectures used in our baseline cGAN model.

The generator is based on the encoder-decoder network with convolutional LSTM (Xingjian et al., 2015). At each step, it takes a frame together with a latent code as inputs and produces the next frame as an output. Contrary to the original SAVP that takes a latent code at each step to encode frame-wise stochasticity, we modified the generator to take one latent code per sequence that encodes the global dynamics of a video. Then the discriminator takes the entire video as an input and produces a prediction on real or fake through 3D convolution operations.

#### D.3.3 EVALUATION METRICS

We provide more details about the evaluation metrics used in our experiment. For each test video, we generate 100 random samples with a length of 28 frames and evaluate the performance based on the following metrics:

- *Diversity*: To measure the degree of diversity of the generated samples, we computed the frame-wise distance between each pair of the generated videos based on Mean Squared Error (MSE). Then we reported the average distance over all pairs as a result.

- $Dist_{min}$: Following Lee et al. (2018), we evaluate the quality of generations by measuring the distance of the closest sample among the all generated ones to the ground-truth. Specifically, for each test video, we computed the minimum distance between the generated samples and the ground-truth based on MSE and reported the average of the distances over the entire test videos.

- $Sim_{max}$: As another measure for the generation quality, we compute the similarity of the closest sample to the ground-truth similar to $Dist_{min}$ but using the cosine similarity of features extracted from VGGNet (Simonyan & Zisserman, 2015). We report the average of the computed similarity for entire test videos.

### D.3.4 MORE EXAMPLES

We present more video prediction results on both BAIR and KTH datasets in Figure H, which corresponds to Figure 6 in the main paper. As discussed in the main paper, the baseline cGAN produces realistic but deterministic outputs, whereas both SAVP and our method generate far more diverse future predictions. SAVP fails to generate the diverse outputs in KTH datasets, mainly because the dataset contains many examples with small motions. On the contrary, our method generates diverse outputs in both datasets, since our regularization directly penalizes the mode-collapsing behavior and force the model to discover various modes. Interestingly, we found that our model sometimes generates actions different from the input video when the motion in input frames are ambiguous (*e.g. hand-clapping* to *hand-waving* in the highlighted example). It shows that our method can generate diverse and meaningful futures.

Figure I presents more detailed qualitative comparison in BAIR robot arm dataset. Both baseline cGAN and SAVP often suffer from the noise predictions in the background, since they fail to predict the correct motion of foreground objects. On the other hand, our method can generate more clear outputs and sometimes even an interaction between foreground and background objects by predicting more meaningful dynamics of videos from latent code $z$. See captions of Figure I for more detailed discussions.

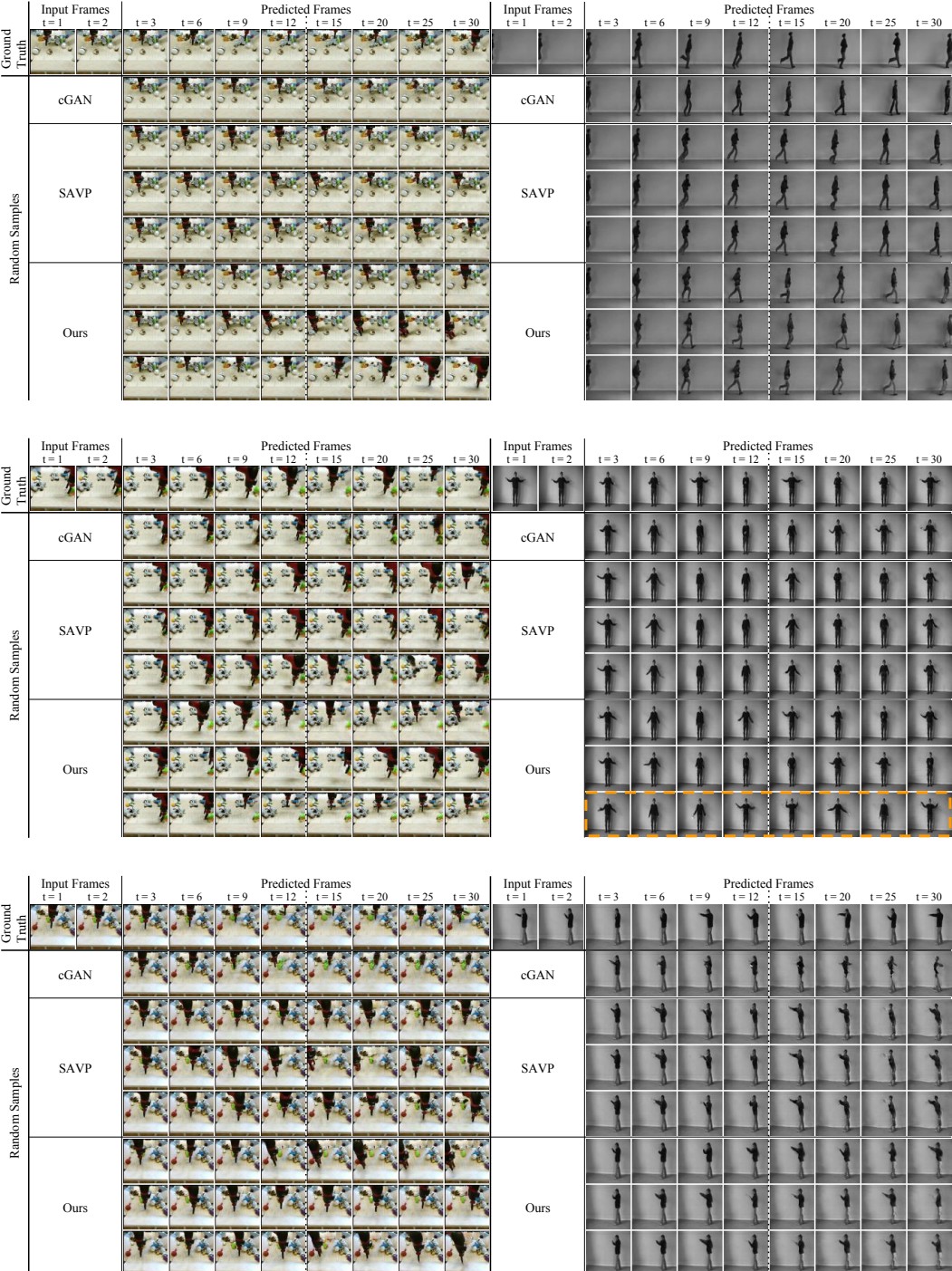

Figure H: Stochastic video prediction results. In both datasets, our method presents diverse prediction, whereas SAVP generate less diverse result especially in the KTH dataset. Interestingly, as you can see from the dotted orange box, our model can explore not only the original condition (hand clapping) but also other cases (hand waving) if the context is not too strong. Please check our web page to see the videos: https://sites.google.com/view/iclr19-dsgan/

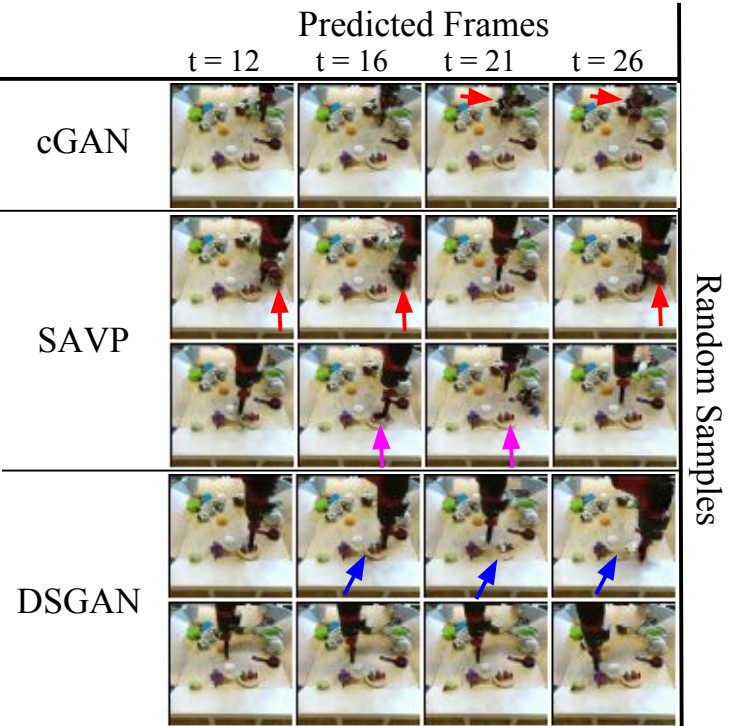

Figure I: Qualitative comparison of various video prediction methods. Both baseline cGAN and SAVP exhibit some noises in the predicted videos due to the failures in separating the moving foreground object from the background clutters (red arrow). Compared to this, our method tends to generate more clear predictions on both foreground and background. Interestingly, SAVP sometimes fail to predict interaction between objects (magenta arrows). For instance, the objects on a table stay in the same position even after pushed by the robot arm. On the other hand, our method is able to capture such interactions more precisely (blue arrows).

