# OpenReview forum: "Diversity-Sensitive Conditional Generative Adversarial Networks"
_ICLR.cc/2019/Conference_

### Official Review · AnonReviewer2 · 2018-11-02
**Interesting idea with good experimental validation**

**Rating:** 7
**Confidence:** 5

**Review:**

The paper proposes a method for generating diverse outputs for various conditional GAN frameworks including image-to-image translation, image-inpainting, and video prediction. The idea is quite simple, simply adding a regularization term so that the output images are sensitive to the input variable that controls the variation of the images. (Note that the variable is not the conditional input to the network.) The paper also shows how the regularization term is related to the gradient penalty term. The most exciting feature about the work is that it can be applied to various conditional synthesis frameworks for various tasks. The paper includes several experiments with comparison to the state-of-the-art. The achieved performance is satisfactory.

To the authors, wondering if the framework is applicable to unconditional GANs.

---

> ### Author Response · Authors · 2018-11-26
> **Response to Reviewer 2**
>
> We appreciate your insightful and constructive comments. We have updated the paper to include the experiments on unconditional GAN (section C.1). Below we provide our response to your comment.
>
> (1). Is the proposed method applicable to unconditional GANs?
>
> We believe that our regularization can be applied to unconditional GAN to relax the mode-collapse problem. To demonstrate this idea, we conducted an experiment using the synthetic data and unconditional GAN model employed in Srivastava et al., 2017. Please see Section C.1 of our revised paper for comprehensive descriptions on experiment settings and results. Below we provide a summary of this experiment.
>
> In this experiment, we used a mixture of eight 2D Gaussian distributions arranged in a ring as a synthetic dataset. We observe that vanilla GAN experiences a severe mode collapse, putting a significant probability mass around a single mode. On the other hand, applying our regularization effectively resolves the mode-collapse problem, enabling the generator to capture all eight modes. Interestingly, our method achieved even higher performance over Srivastava et al., 2017, which also addresses the mode collapse in GAN but for unconditional generation task. It shows that the proposed regularization is also effective in resolving mode collapse problem in unconditional GAN setting.

---

### Official Review · AnonReviewer1 · 2018-11-03
**Well written paper with a simple idea for preventing mode-collapse in GANs but with insufficiently experimental validation**

**Rating:** 6
**Confidence:** 5

**Review:**

The paper proposes a simple way of addressing the issue of mode-collapse by adding a regularisation to force the outputs to be diverse. Specifically, a loss is added that maximises the l2 loss between the images generated, normalised by the distance between the corresponding latent codes. This method is also used to control the balance between visual quality and diversity.

The paper is overall well written, introducing and referencing well existing concepts, and respected the 8 pages recommendation.

Why was the maximum theta (the bound for numerical stability) incorporated in equation 2? What happens if this is omitted in practice? How is this determined?

In section 4, an increase of the gradient norm of the generator is implied: does this have any effect on the robustness/sensitivity of the model to adversarial attacks?

In section 5, how is the “appropriate CGAN” determined?

My main issue is with the experimental setting that is somewhat lacking. The visual quality of the samples illustrated in the paper is inferior to that observed in the state-of-the-art, begging the question of whether this is a tradeoff necessary to obtain better diversity or if it is a consequence of the additional regularisation.. The diversity observed seems to mainly be attributable to colour differences rather than more elaborate differences. Even quantitatively, the proposed method seems only marginally better than other methods.

Update post rebuttal
-----------------------------
The experimental setting that is a little lacking. Qualitatively and quantitatively, the improvements seem marginal, with no significant improvement shown. I would have liked a better study of the tradeoff between visual quality and diversity, if necessary at all.

However, the authors addressed well the issues. Overall, the idea is interesting and simple and, while the paper could be improved with some more work, it would benefit the ICLR readership in its current form, so I would recommend it as a poster -- I am increasing my score to that effect.

---

> ### Author Response · Authors · 2018-11-26
> **Response to  Reviewer 1 (part 1)**
>
> We appreciate your constructive and detailed comments. Due to the character limit of openreview comment system, we provide our responses in two parts.
>
> (1). “Why was the maximum theta (the bound for numerical stability) incorporated in Equation 2? What happens if this is omitted in practice? How is this determined?”
>
> In principle, our regularization term (||G(x,z1)-G(x,z2)||/||z1-z2||) is an unbounded operator because 1) its numerator can grow arbitrarily large with the unbounded generator and 2) its denominator can approach zero with the almost identical latent codes. The \tau in Equation 2 provides a bound to our regularization term thus ensures its numerical stability. However, we found that our regularization term is practically bounded in most conditional GAN implementations because 1) the generator output is usually bounded by non-linear output function (e.g. [0,1] for sigmoid, [-1,1] for the hyperbolic tangent) and 2) it is very unlikely to sample two near-identical latent codes from standard normal distribution. Specifically, we can probably bound the probability of sampling two random codes z and z' from the N-dimensional multivariate standard normal distribution within a distance of \delta by p(|z-z'|<\delta) \leq (\delta/\sqrt(2\pi))^N. For sufficiently small \delta, we can see that such probability decreases exponentially with the size of the latent code. For example, when \delta=0.001 and N=10, this bound is about 10^-30, which implies that the probability that such an event happens is practically zero. (We will include the proof in the next revision. We are happy to provide more details upon request.). For these reasons, we omitted the \tau in Equation 2 in practice as we described in the following sentence of Equation 3. The only hyper-parameter in our formulation (and in all our experiments) is thus \lambda in Equation 3, which controls the importance of the regularization.
>
>
> (2). “In section 5, how is the ‘appropriate CGAN’ determined”?
>
> For each conditional generation task in the experiment, we chose strong cGAN models from the literature that produces realistic but deterministic outputs, as we described in the 4th line of Section 5. We provided details of cGAN baseline in each task in the second paragraph of each corresponding subsection as follows:
>  - Image to image translation (section 5.1): Zhu et al., 2016
>  - Image inpainting (section 5.2): Iizuka et al., 2017
>  - Video prediction (section 5.3): Lee et al., 2018
> These models are considered as among state-of-the-art methods (if not “the state-of-the-art” for each task domain). As we described in the paper, we employed the exact same network architectures and hyperparameters provided by the authors for these models, as our formulation requires modification on only objective function. To be self-contained, please note that we also provided detailed settings of these baseline cGANs in the appendix (Section D.1.1, D.2, and D.3.2).
>
>
> (3). “The visual quality of the samples illustrated in the paper is inferior to that observed in the state-of-the-art …”
>
> Based on our experiment results, we did not observe noticeable quality degradation of our method over its cGAN counterparts. As we discussed in the paper, we conducted a human evaluation study to compare visual realism among cGAN baseline, BicycleGAN, and our method. However, we found that there is no clear winning method over others, which implies that the visual quality of samples is in a similar level for these methods. In terms of FID score, on the other hand, our method consistently achieved substantial improvement over cGAN baseline and BicycleGAN (Table 1 and 3), which shows that the distribution of the generated samples by our method (with improved diversity) matches much better to the true distribution than others.
>
> Please note that our main contribution is improving diversity in existing cGANs, which is orthogonal (and complementary) to achieving high-level visual realism via sophisticated architectural designs or training strategies, e.g., BicycleGAN, pix2pixHD, SAVP, ProgressiveGAN, BigGAN (unpublished concurrent ICLR submission) to name a few. Practically, We showed that, with a few lines of additional code, the diversity of generated samples dramatically improves upon all strong-performing cGAN models we tried. Although an exhaustive demonstration of our regularization to the latest and the most resource-heavy cGAN models (e.g., ProgressiveGAN or BigGAN) was infeasible due to time/resource limit for our submission, we believe our experiments provide compelling evidence of wide applicability of our regularizer.

---

> > ### Author Response · Authors · 2018-11-26
> > **Response to  Reviewer 1 (part 2)**
> >
> > (4). “The diversity observed seems to mainly be attributable to colour differences rather than more elaborate differences”
> >
> > We would like to clarify that the diversity encouraged by our method is not limited to the color differences. In our experiments, we demonstrated more elaborate sample differences on various datasets and tasks as follows:
> >
> > - In map->photo dataset (Figure 2, the last row), our method generates various landmark textures especially on park areas, such as trees, grass, and playgrounds.
> > - In cityscape dataset (Figure 3), our method generates different object textures (buildings, cars, road), lightings and shadows, etc.
> > - In face dataset (Figure 5), our method generates various facial attributes such as gender, age, facial expression, makeups, etc.
> > - In video datasets (Figure 6), our method generates various object dynamics such as different motion categories and speed.
> >
> > Please note that the diversity in the face and video datasets is very subtle in terms of color differences. For instance, we can create different facial expressions or motions by modifying a few pixels around facial or body landmarks. Experiment results show that our method learns semantically more meaningful factors of variations than just color differences. We also remark that our method tends to capture semantically more meaningful diversities than other approaches (i.e. BicycleGAN, SAVP), as shown in Figure B, C, E, and F.
> >
> > Finally, we demonstrate in the paper that we can incorporate an additional encoder into our regularization, which allows us to capture more meaningful sample differences (Equation 6). This can be very useful when the semantic distance of the samples is not well captured in the generator output space (e.g. face, sentence).
> >
> >
> > (5). The proposed method is marginally better than other methods.
> >
> > We believe that our method achieved meaningful improvement over existing state-of-the-art multimodal cGAN methods. Although its performance is competitive with BicycleGAN on some datasets (Table 1), we showed that our method can achieve substantially better performance over a wide range of latent dimensions (Table 2), and more challenging datasets (Table 3). Compared to SAVP, our method achieved substantially diverse and realistic results especially on challenging KTH datasets, where SAVP ends up generating deterministic outputs (Table 5).
> >
> > More importantly, we believe that the main contribution of this paper is proposing a general and principled approach to promote diversity in conditional GANs. Our method achieved consistent improvement over state-of-the-arts on various tasks and frameworks, although such competitors are designed specifically for each task and require non-trivial modifications of baseline cGAN models. As pointed out by other reviewers, alleviating the need to investigate significant changes to model families by focusing instead on a novel optimization objective is an important contribution towards understanding how conditional generative models like cGANs behave.
> >
> >
> > (6). “In section 4, an increase of the gradient norm of the generator is implied: does this have any effect on the robustness/sensitivity of the model to adversarial attacks?”
> >
> > We are sorry, but we could not understand your question clearly. Below we provide a response based on our best guess on your question, but please let us know if it does not address your concern. We are happy to elaborate and discuss further based upon your comments.
> >
> > We assume that your concern is on the sensitivity of the generator against the adversarial perturbation on the input condition x, as our regularization increases the norm of the generator gradient. Denoting the very small perturbation as p, your concern can be rephrased as “Will the proposed regularization increase ||G(x+p,z) - G(x,z)||?”. Our answer is “not necessarily yes”, as our regularization increases the sensitivity of generator over latent code (||G(x,z1) - G(x,z2)||), not an input condition (||G(x+p,z) - G(x,z)||). Also, in another perspective, the cGAN is trained driven with the conditional likelihood (on top of adversarial loss) as a major objective (e.g., “L2 loss” between the generator output and ground-truth output), so a reasonably trained generator model should capture the implicit relationship between input/output pairs in the data and thus would exhibit proper degree of sensitivity to sufficiently different x values (while generating smooth/similar outputs when given very similar x values). Note that we do not need to worry about an adversarial attack on the latent code, as it is always sampled by the model and hidden to users.
> >
> > To be more concrete, we will provide some experimental results if the reviewer can elaborate more on the attack method (e.g. reference). To our best knowledge, we are not aware of any existing works on the adversarial attack against cGAN generator.

---

### Official Review · AnonReviewer3 · 2018-11-03
**An interesting and simple idea.**

**Rating:** 7
**Confidence:** 3

**Review:**

The paper proposes a regularization term for the conditional GAN objective in order to promote diverse multimodal generation and prevent mode collapse. The regularization maximizes a lower bound on the average gradient norm of the generator network as a function of the noise variable.

The regularization is a simple addition to existing conditional GAN models and is certainly simpler than the architectural modifications and optimization tweaks proposed in recent work (BicycleGAN, etc). It is useful to a such a simple solution for preventing mode collapse as well as promoting diversity in generation.

It is shown to promote the generator landscape to be more spread out by lower bounding the expected average gradient norm under the noise distribution. This is a point to be noted when comparing with other work which focus on the vanishing gradients through the discriminator and try to tweak the discriminator gradients. It is a surprising result that such a penalty on the lower bound can prevent mode collapse while also promoting diversity, since I would expect that upper bounding the generator gradient (i.e. lipschitz continuity which wasserstein GANs and related work rely on but for their discriminator instead) makes sense if a smooth interpolation in latent space is desired.

It is also not evident how the vanishing discriminator gradient problem is solved using this regularization -- will it work if the discriminator is allowed to converge before updating the generator?

This simple regularization presented in this paper and its connection to preventing mode collapse feels like an important step towards understanding how conditional generative models like cGANs behave. Alleviating the need to investigate significant changes to model families by focusing instead on a novel optimization objective is an important contribution.

---

> ### Author Response · Authors · 2018-11-26
> **Response to Reviewer 3**
>
> We appreciate your insightful and supportive comments. We have updated the paper to address your concern with converged discriminator (section C.2). Below we provide our responses to your comments.
>
> (1) “I would expect that upper bounding the generator gradient makes sense if a smooth interpolation in latent space is desired”
>
> Our regularization increases the lower-bound of the generator gradient norm to ensure the sensitivity of the generator with respect to the latent code z. Without such bound, we found that the norm of the generator gradient approaches to zero as training progresses, which makes the conditional generator ignoring z.
>
> We can still ensure the smoothness of latent space by bounding our regularization term. This can be achieved implicitly by balancing our regularization with the adversarial loss using the hyperparameter \lambda (Equation 2), or explicitly by introducing an upper-bound to our regularization (\tau in Equation 2). We found that the first trick alone works practically very well to learn smooth latent manifold. (Empirically we found that “no upper bounding” (i.e., \tau=\infty) worked just well; please see our response (1) to Reviewer 1 for more detailed information.) Please see sample interpolation results in Figure D, E, G in appendix and videos on the anonymous website (https://sites.google.com/view/iclr19-dsgan/), where we can observe smooth and continuous transition between samples.
>
>
> (2) “will it work if the discriminator is allowed to converge before updating the generator?”
>
> Thank you for your insightful comment. We empirically validate the effectiveness of our regularization on vanishing gradient problem and reported the results in Section C.2. As suggested by the reviewer, we simulate the vanishing gradient problem by training cGAN baseline until it converges, and retraining the generator from scratch with our regularization while initializing the discriminator with the pre-trained one. Empirically we observed that the pre-trained discriminator can distinguish the real data and generated samples from the randomly initialized generator almost perfectly, and the generator experiences a severe vanishing gradient problem at the beginning of the training. However, even in such cases, we found that the diversity-sensitive regularization helped overcoming this issue throughout the training.
>
> In our experiment on label->image dataset, we found that the generator with our regularizer converges to the similar FID/LPIPS scores (FID: 52.31; LPIPS: 0.16) as the ones reported in the paper (FID: 57.20, LPIPS: 0.18). We observed that our regularization term encourages the generator to efficiently explorer the output space in the early training stage when the discriminator gradients are vanishing, which helps the generator to capture useful gradient signals from the discriminator in the later course of training. This trend can be observed more clearly on our experiments on the synthetic dataset (Section C.1), where our diversity-sensitive regularization spreads the generator landscape and captures meaningful modes. Please find the Section C.2. in the revised paper for more detailed experiment settings and discussions.

---

### Public Comment · ~Augustus_Odena1 · 2018-11-01
**Related work :)**

Hey, I think our ICML paper http://proceedings.mlr.press/v80/odena18a.html qualifies as related work in this case.
In particular, the Jacobian Clamping algorithm from that paper is pretty similar, though we bound the largest and smallest singular values of the generator jacobian and it looks like you approximately maximize the norm of the whole thing.

---

> ### Author Response · Authors · 2018-11-05
> **Thank you for your comment!**
>
> Hi Augustus.
>
> Thank you for your comment. We will definitely include your paper in the revision of our paper as two methods are related. As you mentioned, Jacobian clamping and our regularizer optimize similar but different objective functions (i.e. the former clamps the generator Jacobian within a certain range, while the later increases it with some rough upper-bound), which leads to different impacts on the generator in practice. In our initial attempts, we tried Jacobian clamping on Facade dataset with grid hyper-parameter search but could not achieve the similar FID / LPIPS score as our method. We will add more thorough discussion and investigation results in the revised version of our paper. Thank you.

---

### Meta-Review · Area_Chair1 · 2018-12-14
**a simple regularization for preventing mode collapse**

**Confidence:** 4
**Recommendation:** Accept (Poster)

**Metareview:**

The paper proposes a regularization term on the generator's gradient that increases sensitivity of the generator to the input noise variable in conditional and unconditional Generative Adversarial networks, and results in multimodal predictions. All reviewers agree that this is a simple and useful addition to current GANs. Experiments that demonstrate the trade off between diversity and generation quality would be important to include, as well as the experiment on using the proposed method on unconditional GANs, which was conducted during the discussion period.